# Intertrochanteric Femoral Fractures: A Comparison of Clinical and Radiographic Results with the Proximal Femoral Intramedullary Nail (PROFIN), the Anti-Rotation Proximal Femoral Nail (A-PFN), and the InterTAN Nail

**DOI:** 10.3390/medicina59030559

**Published:** 2023-03-13

**Authors:** Mustafa Yalın, Fatih Golgelioglu, Sefa Key

**Affiliations:** 1Department of Orthopedics and Traumatology, Fethi Sekin City Hospital, Elazığ 23280, Turkey; fatihgolgelioglu@gmail.com; 2Department of Orthopedics and Traumatology, Faculty of Medicine, Firat University, Elazığ 23190, Turkey; sefa_key@hotmail.com

**Keywords:** intertrochanteric femoral fractures, proximal femoral nail, Z-effect, V-effect

## Abstract

*Background and Objectives:* The aim of this study was to evaluate retrospectively the radiological and functional outcomes of closed reduction and internal fixation for intertrochanteric femoral fractures (IFF) using three different proximal femoral nails (PFN). *Materials and Methods:* In total, 309 individuals (143 males and 166 females) who underwent surgery for IFF using a PFN between January 2018 and January 2021 were included in the study. Our surgical team conducted osteosynthesis using the A-PFN^®^ (TST, Istanbul, Turkey) nail, the PROFIN^®^ (TST, Istanbul, Turkey), and the Trigen InterTAN (Smith & Nephew, Memphis, TN, USA) nail. The PFNs were compared based on age, gender, body mass index (BMI), length of stay (LOS) in intensive care, whether to be admitted to intensive care, mortality in the first year, amount of transfusion, preoperative time to surgery, hospitalisation time, duration of surgery and fluoroscopy, fracture type and reduction quality, complication ratio, and clinical and radiological outcomes. The patients’ function was measured with the Harris Hip Score (HHS) and the Katz Index of Independence in Activities of Daily Living (ADL). *Results:* Pain in the hip and thigh is the most common complication, followed by the V-effect. The Z-effect was seen in 5.7% of PROFIN patients. A-PFN was shown to have longer surgical and fluoroscopy durations, lower HHS values, and much lower Katz ADL Index values compared to the other two PFNs. The V-effect occurrence was significantly higher in the A-PFN group (36.7%) than in the InterTAN group. The V-effect was seen in 33.1% of 31A2-type fractures but in none of the 31A3-type fractures. *Conclusions:* InterTAN nails are the best choice for IFFs because they have high clinical scores after surgery, there is no chance of Z-effect, and the rate of V-effect is low.

## 1. Introduction

Over a million individuals suffer from hip fractures every year, making it one of the most prevalent orthopaedic injuries [1,2]. Nearly half of all hip fractures are intertrochanteric femur fractures (IFFs), which are more common in the elderly and are often the consequence of low-energy traumas [3]. In order to achieve a satisfactory reduction and to facilitate the patients’ early recovery, surgical treatment is required for such kinds of fractures [4]. The dynamic hip screw (DHS) implant, which was previously regarded as the gold standard therapy for stable intertrochanteric fractures, was shown to be inadequate for the stabilisation of fractures of the unstable kind [5]. As opposed to extramedullary devices such as DHS, proximal femoral nails (PFN) have a biomechanical advantage because of their closer location to the vector of force line and shorter moment arm [6]. Moreover, based on the results of several reports, intramedullary fixation may be preferable to extramedullary fixation for patients since there is a lower risk of implant failure and reoperation, and functional scores are higher [7,8,9]. It is possible to implant a PFN with a minimally invasive procedure. By performing a closed reduction of the fracture, the haematoma is maintained, and the surgeon can do a minimally invasive procedure with minimum soft-tissue dissection, thereby minimizing surgical trauma, blood loss, infection, and wound complications [10,11].

Bone healing is significantly aided by the interfragmentary linear compression provided by the lag screw in the majority of PFNs. One or two lag screws, integrated or locked lag screws, and a wedge block that offers rotational stability are some of the PFN designs on the industry [12]. Clinical and radiological evaluations of two distinct PFNs have been the subject of several published papers [12,13,14,15]. Even though PFNs such as A-PFN, PROFIN, and InterTAN are widely used in our country, very few studies have compared the radiological and clinical results of these three PFNs. Hence, the aim of the current study was to compare three distinct PFNs in terms of radiological and functional results in patients treated with closed reduction and internal fixation for IFFs.

## 2. Materials and Methods

A retrospective observational study was conducted, which included the evaluation of clinical data. After receiving approval from the local ethics committee (22 May 2021), patients who underwent surgery for IFFs between January 2018 and January 2021 were reviewed. All patients provided a written informed consent form in compliance with the hospital’s ethical committee’s norms. Individuals with unilateral isolated IFF who were mobile enough to undertake everyday tasks prior to the injury, were at least 18 years old, and whose follow-up period was at least 1 year participated in the study. Non-trochanteric fractures, pathologic fractures, polytrauma patients, bilateral simultaneous fractures, previous intertrochanteric fractures in the contralateral leg, impaired muscle strength, mental disorders including dementia and follow-up less than a year were the exclusion criteria. A total of 309 individuals (143 males and 166 females) who met the inclusion criteria were included in the study. Preoperative pelvic or hip images were employed to categorise fractures using the AO (Arbeitsgemeinschaft für Osteosynthesefragen) classification system. Surgeons have implanted three distinct PFNs, comprising InterTAN, PROFIN, and A-PFN (Figure 1). The PFNs were contrasted on several variables, including age, gender, body mass index (BMI), energy of trauma, length of staying (LOS) in intensive care, whether to be admitted to intensive care, mortality in the first year, amount of transfusion, preoperative time to surgery, hospitalisation time, duration of surgery and fluoroscopy, fracture type and reduction quality, complication ratio, and clinical and radiological outcomes. The functionality of patients was determined with the Harris Hip Score (HHS) and the Katz Index of Independence in Activities of Daily Living (ADL). For evaluating a person’s ADL, a modified version of Katz’s questionnaire [16] was used, which included six questions regarding self-care and four questions related to mobility. Respondents who were not capable of performing an activity without assistance or without significant trouble were labelled as limited regarding that activity. The number of limits was then written down, and a score between 0 and 10 was given. The reliability and validity of self-reported limitations on the Katz ADL were also investigated by Reijneveld et al. [17], who found that, in terms of clinical care, their findings support the idea that assessing self-reported ADL provides an acceptable, reliable, and valid measure of functional status. An experienced orthopaedic surgeon (SK) who was not participating in the patient’s care reviewed the patient’s radiological outcomes, including reduction quality, fracture union, and radiological complications including V-effect, Z-effect, and varus collapse. Postoperative reduction quality was evaluated by utilising Baumgartner reduction metrics, as revised by Fogagnolo et al. [18].

The kind of implant we employed for a patient was governed by the hospital’s budget and the Ministry of Health’s purchasing rules, both of which have changed over time. Hospital administration was charged for the type of nail. Neither doctors nor patients were responsible for this decision and were aware of the type of nail used just before the operation.

### 2.1. Characteristics of PFNs

#### 2.1.1. InterTAN (Intertrochanteric Antegrade Nail)

The titanium alloy used in the production of InterTAN PFN allows for a proximal 4° valgus offset. The nail features a trapezoidal cross-section with a 17 mm proximal diameter and a 10 to 11.5 mm grooved distal tip diameter. InterTAN PFNs are available with either a 125° or 130° collodiaphyseal angle (CDA). A lag screw of 11 mm and a compression screw of 7 mm were used. The tip of the nail was secured by a single screw that was locked in either a dynamic or static configuration. With the combined proximal screw system, it was possible to achieve interfragmentary compression of up to 15 mm. The InterTAN nail is designed with an interlocking lag nail system that helps minimise femoral head movement and prevents the femoral head from collapsing [7].

#### 2.1.2. A-PFN (Antirotational Proximal Femoral Nail)

A-PFN (TST Medical Devices, Istanbul, Turkey^®^) is available in two different lengths: 160 and 220 mm. The top portion of the proximal nail features a 6° valgus angle (mediolateral curvature) with a diameter of 15 mm. This nail is available in four different diameters: 9, 10, 11, and 12; it has a lag screw that compresses the fractures and a wedge block that provides rotational stability for femoral fractures. The 10 mm wide thread and 125° angle of the lag screw are consistent with the CDA. The wedge block is in the groove on the lower portion of the lag screw. The distal end of the nail has two locking holes appropriate for either dynamic or static fixations, as well as a slot [19].

#### 2.1.3. PROFIN (Proximal Femoral Intramedullary Nail)

PROFIN PFN is a titanium alloy tube with a cannulated and flat design. It features a proximal valgus offset of 6° and a distal grooved shape, and it is attached with two 8.5 mm lag screws with 135° CDA. The surgical compression of interfragmentary fractures was also possible with this system. The nail has a 16 mm diameter at its proximal end and three separate distal diameters measuring 10, 11, and 12 mm. Both dynamic and static fixation with 4.5 mm locking screws are possible via the two distal holes [20].

### 2.2. Surgical Procedure

Every patient received 1 g of parenteral cefuroxime sodium 60 min prior to surgical incision. All procedures were carried out under general or regional anaesthesia. All procedures were performed by the same surgical team consisting of five surgeons with five years of expertise in orthopaedic trauma, with the patient laying supine on a traction table after a closed reduction under fluoroscopic guidance. Nailing was performed by utilizing a minimally invasive technique after a closed reduction was achieved under fluoroscopic control. Each of the three PFN kinds was inserted through the trochanter major. By installing the lag screw in InterTAN and employing the integrated compression screw, interfragmentary compression was achieved. Interfragmentary compression was accomplished in PROFIN by inserting two different lag screws into the nail. A lag screw and wedge block placed through the nail were used to generate interfragmentary compression in the A-PFN. Our surgical team performed osteosynthesis with the A-PFN^®^ (TST, Istanbul, Turkey) nail, the PROFIN^®^ (TST, Istanbul, Turkey), and the Trigen InterTAN (Smith & Nephew, Memphis, TN, USA) nail. All three PFNs likewise had the distal hole statically locked. In order to prevent venous thromboembolism (VTE), low molecular weight heparin was administered to all patients when they were admitted to the hospital. VTE prophylaxis administration was stopped twelve hours before the procedure, and then it was restarted six hours following the operation. All individuals were provided with precisely the same postoperative care. At 24 h postoperatively, all patients were given 4 × 1 gramme of cefazolin sodium intravenously as prophylaxis. Each patient was given enoxaparin for 14 days postoperatively to prevent thromboembolism, with the dosage based on their body mass index. Patients were encouraged to walk with a walker and start weight bearing as tolerated on the first postoperative day, when they also began quadricep exercises. Individuals were recommended to begin partial weight bearing two weeks following the procedure. A physiotherapist who also worked in the intensive care unit (ICU) helped give passive range-of-motion exercises to ICU patients while they were lying in bed. Walking and range-of-motion exercises were maintained after patients were admitted to the clinic. Upon radiological confirmation of fracture healing, patients were given clearance to begin full weight bearing. The duration of the surgery was determined to be the time from the first incision made on the patient following closed reduction of the fracture and the complete closure of the wound. Fluoroscopy time was calculated based on the total number of exposures taken at the conclusion of the procedure. Healing of a bone fracture was described as the development of cortical integrity, including at least three cortices or a bridging callus.

Clinical and radiographic assessments were performed on all patients by the same surgical team two weeks, three months, and six months following surgery, and once yearly afterwards. Hip and thigh pain was defined as a mild pain that responded to paracetamol treatment. It was considered a complication if it persisted at the third-month follow-up and was questioned by the same surgical team at each follow-up. Severe pain that did not respond to painkillers was investigated in terms of implant failure. Using the CDA of the patient’s contralateral hip, the degree of varus collapse was determined. When the same surgical team that conducted the operation encountered complications during the postoperative follow-up, they advised further surgery to the patients if considered reasonable.

### 2.3. Statistics

For statistical purposes, IBM SPSS Statistics 22 (IBM SPSS, Turkey) was used to evaluate the study’s findings. Using the Shapiro–Wilk test, the conformance of the study’s parameters to the normal distribution was determined while the data were analysed. The results of the Shapiro–Wilk test as well as the histogram graphics and boxplot findings were considered while it was decided which test to apply. Because of their length, histogram and boxplot findings are not included in the final draft. In addition to descriptive statistical methods (mean, standard deviation, and frequency), the Kruskal–Wallis test was used to compare parameters that did not exhibit normal distribution in the comparison of quantitative data, and the Dunn’s test was used to identify the group responsible for the difference. Using the chi-square test, the Fisher–Freeman–Halton test, and the continuity (Yates) correction, qualitative data were compared. Significance was evaluated at the *p* < 0.05 level.

## 3. Results

The study was conducted on a total of 309 cases—143 (46.3%) men and 166 (53.7%) women—aged between 23 and 95 between January 2018 and January 2021. The Shapiro–Wilk test, which was used to determine whether the parameters of the study matched a normal distribution, provided the findings shown in Table 1. The mean age of the patients was 77.34 ± 7.99 years. Table 2 and Table 3 present the descriptive features of the patients. Simple falls are the most prevalent cause of injury across all categories. The average time between admission and surgery was 2.26 days. The average length of time between surgery and discharge was 4.95 days. The mean duration of surgery was 65.45 min, and the 1-year mortality rate was 27.5%. In 66.3% of the patients, anatomical reduction was achievable, whereas 7.8% of the reductions were of poor quality. Approximately 39.5% of patients had at least one complication, with 31-A2 fractures being the most prevalent kind of fracture.

A-PFN was shown to have longer surgical and fluoroscopy durations compared to the other two PFNs (Table 4). Based on the post hoc evaluations, A-PFN had considerably lower HHS values compared to the other two PFNs (Table 5). Comparing the Katz ADL Index values revealed a substantial difference between A-PFN and InterTAN, with A-PFN having much lower Katz ADL Index values compared to InterTAN (Table 5).

Hip and thigh pain, in particular, stands out as the most common complication. Hip and thigh pain was reported by 29 individuals in PROFIN, 22 in InterTan, and 28 in A-PFN. The V-effect ranked as the second most frequent complication. The V-effect was detected by 9 individuals in PROFIN, 3 in InterTAN, and 18 in A-PFN. We detected the Z-effect in 6 of 104 patients treated with PROFIN. In one patient each, hardware breakage, double screw-back, and non-union were detected (Table 6).

The incidence of the V-effect differed significantly (*p*:0.022; *p* < 0.05) depending on the PFN subtype (Table 6). Pairwise comparisons revealed that the incidence of the V-effect in the A-PFN group (36.7%) was significantly higher than in the InterTAN group (10%) (*p* < 0.05) (Table 7). The Z-effect could not be statistically evaluated among the PFN subgroups since it is just a potential complication in the PROFIN group.

## 4. Discussion

A noteworthy aspect of the current study is the comparison of the three most prevalent PFN models that have been used in our country in recent times. Other notable aspects of the current study also include the complication rates and the association of complications with implant type and fracture type.

The literature reports failure rates of up to 56%, depending on fracture severity and implant design, despite the improvements in operational treatment [21]. Pain, immobility, and the need for further surgery may all result from a failed fixation [22]. Unstable fractures (AO OTA 31-A2 and A3 type) which are multi-fragmentary or have a displaced femoral neck may be especially difficult to fix [23]. Even after the fracture has been reduced and stabilised, there is still a greater risk of failure, especially varus collapse, which may cause pain, functional impairments, and implant failure [24,25].

Yaozeng et al. [26] demonstrated that 90.1% of individuals experienced hip and thigh pain due to the gluteus medius muscle being scraped following nail insertion. However, Kumbaracı et al. [13] discovered in their study that, even though 72% of patients had thigh or hip pain, it had no impact on functional results. Quartley et al. conducted a meta-analysis in 2022 comparing the InterTAN implant to different existing nails for unstable fractures and concluded that the InterTAN nail reduced implant-related failure, re-intervention rates, and hip and thigh pain without impacting recovery results [27]. The current study did not reveal any significant difference in hip and thigh pain between implant types. In their 2019 study, Duramaz et al. [28] reported a hip and thigh pain prevalence of 21.8%, which was similar to the current study’s prevalence of 25.5%, and this prevalence seemed to have no influence on functional or radiological outcomes either.

Employing PFN with two independent lag screws to treat IFFs sometimes leads to issues such as Z-effect or reverse Z-effect [29]. There was a total of six individuals who suffered the Z-effect following PROFIN; however, no patients experienced the reverse Z-effect. The InterTAN nail is designed to improve rotational stability for hip fracture patients by using two interlocking lag screws (not independent) near the proximal end to construct a locking mechanism [27]. It is possible that this might help prevent problems such as femoral neck erosion, varus collapse, and unnatural shortening [30,31]. The current study’s findings showing that InterTAN-treated individuals did not experience the Z-effect are in line with the existing literature. Ertürer et al. [29] claimed in 2012 that the Z-effect may be avoided by using two screws of equal size to distribute the strains on the hip. All patients in the current study who had PROFIN had a longer superior lag screw than an inferior lag screw, and the Z-effect occurred in 6% of individuals in the current study. Investigation into the biomechanics of this theory is clearly required.

The average surgery duration with the A-PFN nail was 55.19 ± 15.51 min, according to the 2018 study by Karakuş et al. [19]. Lin et al. [32] reported a mean operative time of 78.5 min in their study of 231 patients. In the current study, the mean duration of surgery following the application of an A-PFN nail was 74.11 ± 4.19 min, which was longer than that recorded in the literature. We believe that this is due to the fact that our surgeons had challenges during the insertion of the compression screws owing to technical issues with the implant set, resulting in an increase in the frequency and duration of fluoroscopy. Based on data from 2019, Duramaz et al. [28] found that InterTAN surgeries took an average of 61.6 min, while PROFIN surgeries took an average of 64.6 min. The present analysis confirmed the literature-reported values of 60.4 min for InterTAN and 61.2 min for PROFIN.

HHS was the most reported functional outcome for patients with intertrochanteric fractures, as reported in multiple studies [33,34,35,36,37,38,39], and in these reports, no statistically significant differences in HHS were found among PFN groups. Similar to the current study, a 2015 study by Uzer et al. [12] compared HHS values between patients who had InterTAN and PROFIN nails and found no significant difference. The mean HHS levels were also quite similar to those seen in the current study. Compared to InterTAN and PROFIN, A-PFN patients had significantly lower HHS and Katz ADL values in the current study. In their 2018 study, Karakuş et al. [19] evaluated their patients in groups based on age; the mobility scores of patients who received A-PFN were found to be in line with the Katz ADL values of patients in the current study. Evaluating the complication rates of PFNs reveals a statistically significant difference only in terms of the V-effect. Some of the reasons for the poor clinical results in A-PFN patients may be due to the prolonged duration of the surgery, in our perspective.

When the fracture line extends all the way to the greater trochanteric tip, an iatrogenic complication known as the ‘‘V-effect’’ might occur following fixation with PFN in IFFs (Figure 2). Hu et al. [40] were the first to provide a detailed description of the V-effect in the academic literature. The V-effect might be regarded as the result of two mechanisms. The first component is the placement of the guide wire to the fracture line, which continues towards the trochanteric point, instead of the guide wire’s point of entry in accordance with the PFN model. Second, since the guide wire penetrates the intramedullary area in the incorrect way, the drilling performed on it produces a hinge impact rather than an intramedullary hole for PFN [41]. The varus of the femoral neck with respect to the femoral shaft seems to be a direct result of the V-effect. This is related to a large extent to the fact that PFN generates separation in the fracture zone, which eventually reaches the apex of the great trochanter. Based on the results of the current study, 9.7% of participants experienced the V-effect. This was consistent with the 9.4% incidence of the V-effect reported by Eceviz et al. [41] in their 2021 study. They also discovered that InterTAN nails had the lowest occurrence of the V-effect compared to PROFIN nails. Consistent with the previous research, the present study found that the V-effect was most prevalent in the A-PFN group and least prevalent in the InterTAN group. As the second most common complication in the current study, the V-effect was found in 32.9% of 31A2-type fractures but in none of the 31A3-type fractures, leading us to conclude that surgeons should be careful when operating on 31A2-type fractures. Hu et al. [40] advised having an assistant hold the greater trochanter while the surgeon performs reaming at a high rotational speed. Consequently, the proximal femur may have to be widened so that the PFN may be implanted at a more advantageous angle. It was concluded that the V-effect, which contributes to hip varus deformity and non-union, was avoidable. There is a lack of data about the clinical outcomes of patients diagnosed with the V-effect. Several individuals having perfect images did not recover to their former well-being level, while other individuals with the V-effect on images had no noticeable symptoms. Not all of our patients with the V-effect had varus collapse. Clinical results may be affected by the V-effect; hence, further long-term follow-up clinical studies evaluating varus progression in patients with the V-effect and examining this phenomenon are needed.

Several limitations exist in the current study. First, it is a retrospective study restricted to a single tertiary institution in our country; therefore, it has intrinsic drawbacks. Second, the most significant drawback of the research is the wide age range of individuals with hip fractures evaluated. Third, bone mineral density examinations, which could be beneficial in determining the durability of PFNs versus bone density scores, were also not regularly conducted for individuals with hip fractures. Fourth, the study did not identify postoperative complication treatment or surgical techniques. The strength of the study lies in its assessment of the clinical and radiological outcomes of three commonly applied PFN types in our country. The study’s other strengths are its large sample size, its analysis of variations in complication occurrence among implant types, and its contribution to the literature on the V-effect, a rarely mentioned complication that surgeons may encounter frequently.

## 5. Conclusions

Consequently, when the three implant types are compared based on their complication rates and clinical outcomes, the InterTAN nail comes out on top. We consider that InterTAN nails will be an ideal choice for femur fractures due to their relatively short surgery and fluoroscopy times, high postoperative clinical scores, lack of Z-effect probability, and low V-effect rates, which were mentioned rarely in the literature. Multi-centred prospective studies comparing the three kinds of nails with larger patient populations are needed.

## Figures and Tables

**Figure 1 medicina-59-00559-f001:**
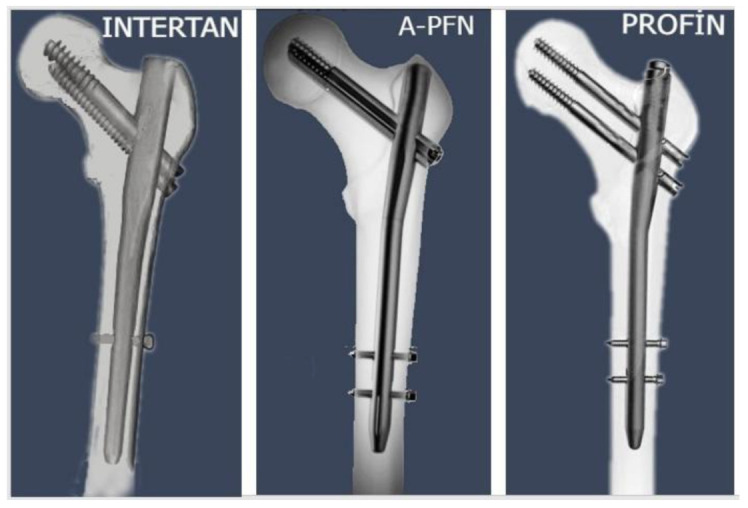
The medical images demonstrate the structural system of the three proximal femoral nails.

**Figure 2 medicina-59-00559-f002:**
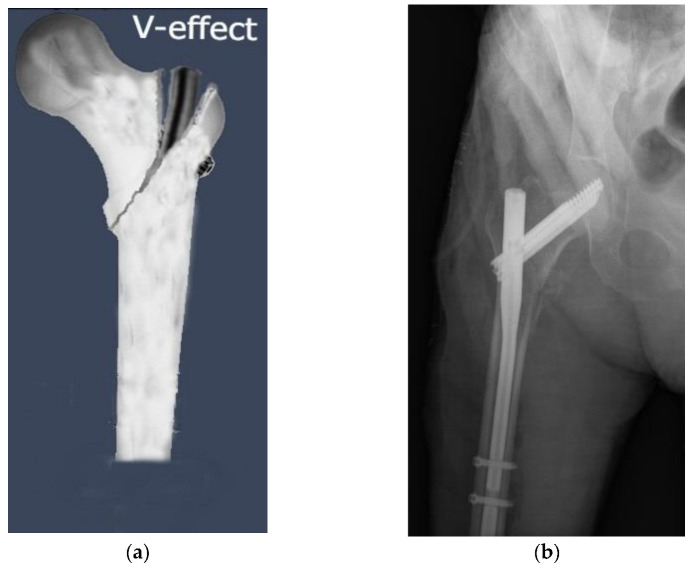
The ‘’V-effect’’, an iatrogenic complication of proximal femoral nail fixation in intertrochanteric femoral fractures, may develop when the fracture line extends all the way to the greater trochanteric tip. (**a**) An illustration of the “V-effect”. (**b**) Observation of the V-effect in a patient who underwent surgery with A-PFN nail.

**Table 1 medicina-59-00559-t001:** Normality test results according to type of nail groups and in total.

	Type of PFN	Shapiro–Wilk-Tests of Normality-*p*
Age	A-PFN	0.392
	InterTAN	0.066
	PROFIN	0.633
	Total	0.026 *
BMI	A-PFN	0.023 *
	InterTAN	0.006 *
	PROFIN	0.032 *
	Total	0.001 *
ASA score	A-PFN	0.000 *
	InterTAN	0.000 *
	PROFIN	0.000 *
	Total	0.000 *
Time from admission to surgery (days)	A-PFN	0.000 *
	InterTAN	0.000 *
	PROFIN	0.000 *
	Total	0.000 *
İntensive care stay (days)	A-PFN	0.000 *
	InterTAN	0.000 *
	PROFIN	0.001 *
	Total	0.000 *
Red blood cell transfusion (units)	A-PFN	0.000 *
	InterTAN	0.000 *
	PROFIN	0.000 *
	Total	0.000 *
Time from surgery to discharge (days)	A-PFN	0.003 *
	InterTAN	0.000 *
	PROFIN	0.000 *
	Total	0.000 *
Duration of surgery (min)	A-PFN	0.020 *
	InterTAN	0.019 *
	PROFIN	0.042 *
	Total	0.000 *
Fluoroscopy time (min)	A-PFN	0.170
	InterTAN	0.028 *
	PROFIN	0.489
	Total	0.000 *

* *p* < 0.05.

**Table 2 medicina-59-00559-t002:** The descriptive features of the patients.

		Min–Max	Mean ± SD (Median)
Age		23–95	77.34 ± 7.99 (76)
Body mass index		23–32	26.35 ± 1.91 (26)
İntensive care stay (days)		1–12	2.95 ± 1.51 (3)
		*n*	%
Gender	Male	143	46.3
	Female	166	53.7
İntensive care	Absence	211	68.3
	Presence	98	31.7
Mortality (in 1 year)	Absence	224	72.5
	Presence	85	27.5
ASA score		1–4	2.74 ± 0.86 (3)
Time from admission to surgery (days)		1–5	2.26 ± 1.56 (2)
Red blood cell transfusion (units)		0–3	0.29 ± 0.48 (0)
Time from surgery to discharge (days)		1–15	4.95 ± 1.87 (5)
Duration of surgery (min)		50–90	65.45 ± 7.83 (64)
Fluoroscopy time (min)		27–65	40.76 ± 8.3 (38)
Harris Hip Score		33–97	72.15 ± 13.89 (75)
Katz ADL Index		1–6	4.2 ± 1.12 (4)

**Table 3 medicina-59-00559-t003:** The descriptive features of the patients.

		*n*	%
Type of PFN	A-PFN	107	34.6
	InterTAN	98	31.7
	PROFIN	104	33.7
Energy of trauma	Simple fall	256	82.8
	Traffic accident	25	8.1
	Falling from high	28	9.1
Fracture side	Right	116	37.5
	Left	193	62.5
Reduction quality	Acceptable	80	25.9
	Anatomic	205	66.3
	Poor	24	7.8
Fracture classification (AO)	31A1	104	33.7
	31A2	179	57.9
	31A3	26	8.4
Anaesthesia type	General	62	20.1
	Spinal	247	79.9
Presence of complications	Absent	187	60.5
	Present	122	39.5
Complications (*n* = 122)	Cut-Out	10	8.2
	Hip and Thigh Pain	79	64.8
	Deep tissue infection	4	3.3
	Double screw back	1	0.8
	Hardware breakage	1	0.8
	Superficial infection	9	7.4
	Varus Collapse	14	11.5
	Urinary System infection	6	4.9
	V-effect	30	24.6
	Z-effect	6	4.9
	Non-union	1	0.8

**Table 4 medicina-59-00559-t004:** Evaluation of general characteristics among proximal femoral nail subgroups.

		Type of PFN			*p*
		A-PFN	InterTAN	PROFIN	
		(Min–Max)–(Mean ± SD (Median))	(Min–Max)–(Mean ± SD (Median))	(Min–Max)–(Mean ± SD (Median))	
Age		(65–92)–(77.81 ± 6.64 (76))	(23–95)–(76.6 ± 9.64 (76))	(56–95)–(77.55 ± 7.57 (77.5))	^1^ 0.799
Body mass index		(24–30)–(26.45 ± 1.68 (26))	(23–32)–(26.29 ± 1.93 (26))	(23–32)–(26.32 ± 2.12 (26))	^1^ 0.521
Intensive care stay (days)		(1–5)–(2.66 ± 1 (2))	(1–12)–(3.52 ± 2.06 (3))	(1–6)–(2.81 ± 1.35 (2))	^1^ 0.101
ASA score		(1–4)–(2.83 ± 0.69 (3))	(1–4)–(2.65 ± 0.93 (3))	(1–4)–(2.73 ± 0.95 (3))	^1^ 0.405
Time from admission to surgery(days)		(1–5)–(2.25 ± 1.61 (1))	(1–5)–(2.32 ± 1.55 (2))	(1–5)–(2.21 ± 1.52 (2))	^1^ 0.777
Red blood cell transfusion (units)		(0–1)–(0.32 ± 0.47 (0))	(0–3)–(0.31 ± 0.55 (0))	(0–1)–(0.26 ± 0.44 (0))	^1^ 0.645
Time from surgery to discharge (days)		(2–7)–(4.76 ± 1.12 (5))	(2–15)–(5.1 ± 2.34 (5))	(1–13)–(5 ± 1.98 (5))	^1^ 0.970
Duration of surgery (mins)		(64–90)–(74.11 ± 4.19 (74))	(50–71)–(60.46 ± 4.67 (60))	(50–85)–(61.24 ± 5.02 (60.5))	^1^ 0.000 *
Fluoroscopy time (mins)		(40–65)–(50.55 ± 4.33 (50))	(27–49)–(35.15 ± 4.08 (36))	(27–47)–(35.97 ± 4.3 (36))	^1^ 0.000 *
Harris Hip Score		(35–94)–(70.84 ± 13.59 (71))	(33–95)–(75.5 ± 13.42 (78))	(37–97)–(70.33 ± 14.19 (75))	^1^ 0.010 *
Katz ADL Index		(2–6)–(4.02 ± 1.08 (4))	(1–6)–(4.45 ± 1.15 (4))	(2–6)–(4.16 ± 1.1 (4))	^1^ 0.020 *
		*n* (%)	*n* (%)	*n* (%)	
Gender	Male	49 (%45.8)	47 (%48)	47 (%45.2)	^2^ 0.918
	Female	58 (%54.2)	51 (%52)	57 (%54.8)	
İntensive care	Absent	72 (%67.3)	71 (%72.4)	68 (%65.4)	^2^ 0.539
	Present	35 (%32.7)	27 (%27.6)	36 (%34.6)	
Mortality (in 1 year)	Absent	73 (%68.2)	75 (%76.5)	76 (%73.1)	^2^ 0.407
	Present	34 (%31.8)	23 (%23.5)	28 (%26.9)	
Energy of trauma	Simple fall	89 (%83.2)	80 (%81.6)	87 (%83.7)	^2^ 0.912
	Traffic accident	8 (%7.5)	10 (%10.2)	7 (%6.7)	
	Falling from high	10 (%9.3)	8 (%8.2)	10 (%9.6)	
Fracture side	Right	42 (%39.3)	35 (%35.7)	39 (%37.5)	^2^ 0.872
	Left	65 (%60.7)	63 (%64.3)	65 (%62.5)	
Reduction quality	Acceptable	27 (%25.2)	21 (%21.4)	32 (%30.8)	^2^ 0.088
	Anatomic	70 (%65.4)	74 (%75.5)	61 (%58.7)	
	Poor	10 (%9.3)	3 (%3.1)	11 (%10.6)	
Anaesthesia type	General	18 (%16.8)	21 (%21.4)	23 (%22.1)	^2^ 0.581
	Spinal	89 (%83.2)	77 (%78.6)	81 (%77.9)	

^1^ Kruskal–Wallis test. ^2^ Chi-square test. ** p* < 0.05.

**Table 5 medicina-59-00559-t005:** Post hoc evaluations of proximal femoral nail subtypes.

	A-PFN-InterTAN	A-PFN-PROFIN	InterTAN-PROFIN
Duration of surgery (min)	0.000 *	0.000 *	0.492
Fluoroscopy time (min)	0.000 *	0.000 *	0.483
Harris Hip Score	0.031 *	0.019 *	0.846
Katz ADL Index	0.006 *	0.308	0.080
ASA Score	0.008 *	0.014 *	0.589
Fracture classification (AO)	0.014 *	0.000 *	0.023 *

* *p* < 0.05.

**Table 6 medicina-59-00559-t006:** Evaluation of complication occurrence among proximal femoral nail subgroups.

		Type of Proximal Femoral Nail			*p*
		A-PFN	InterTAN	PROFIN	
		*n* (%)	*n* (%)	*n* (%)	
Presence of complications	Absent	58 (%54.2)	68 (%69.4)	61 (%58.7)	^1^ 0.076
	Present	49 (%45.8)	30 (%30.6)	43 (%41.3)	
Complications	Cut-Out	3 (%6.1)	3 (%10)	4 (%9.3)	^2^ 0.766
	Hip and thigh pain	28 (%57.1)	22 (%73.3)	29 (%67.4)	^1^ 0.309
	Deep tissue infection	2 (%4.1)	1 (%3.3)	1 (%2.3)	-
	Double screw back	0 (%0)	0 (%0)	1 (%2.3)	-
	Hardware breakage	0 (%0)	0 (%0)	1 (%2.3)	-
	Superficial infection	3 (%6.1)	5 (%16.7)	1 (%2.3)	-
	Varus collapse	7 (%14.3)	2 (%6.7)	5 (%11.6)	^2^ 0.673
	Urinary system infection	1 (%2)	2 (%6.7)	3 (%7)	-
	V-effect	18 (%36.7)	3 (%10)	9 (%20.9)	^1^ 0.022 *
	Z-effect	0 (%0)	0 (%0)	6 (%14)	-
	Non-union	0 (%0)	1 (%3.3)	0 (%0)	-

^1^ Chi-square test. ^2^ Fisher–Freeman–Halton test. * *p* < 0.05.

**Table 7 medicina-59-00559-t007:** Post hoc evaluations of the V-effect among nail subtypes.

	A-PFN-InterTAN	A-PFN-PROFIN	InterTAN-PROFIN
V-effect	0.019 *	0.152	0.180

* *p* < 0.05.

## Data Availability

Not applicable.

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
