# Peer review of "Intertrochanteric Femoral Fractures: A Comparison of Clinical and Radiographic Results with the Proximal Femoral Intramedullary Nail (PROFIN), the Anti-Rotation Proximal Femoral Nail (A-PFN), and the InterTAN Nail"

_medicina, 2023, doi:10.3390/medicina59030559_

Round 1

Reviewer 1 Report

Dear Authors, 

In my opinion this research is suitable for publishing in our journal.

Kinds regards

Author Response

 Thank you for the comments.

Reviewer 2 Report

Thank you for your hard work in organizing a large number of cases. However, some modifications are required.

1. Lines 42 through 46 highlight only some of the recent papers comparing various DHS vs PFNs. Recently, PFN has been reported to have better results than DHS in various ways, and DHS is gradually disappearing around the world. In order to help readers understand and reduce misunderstandings, it is recommended that the entire sentence be modified to fit the latest trend. References should also be rewritten to suit the strengths of the PFN.

2. Classification of study subjects in lines 62-67. The younger age group of the study subjects is included. It seems problematic to put young fractures and osteoporotic fractures in the same category and compare them. It seems necessary to investigate by dividing age or distinguish by BMD.

3. After "Surgical procedure", describe how you planned your follow-up after surgery.

4. The "V-effect" in line 146, 231-241, is a problem caused by insufficient expansion to the medial side before nail insertion. Could this be a technical problem during surgery? Please describe this point.

5. Was the operation performed by a single, skilled surgeon? Or was it performed by multiple surgeons? Please mention this earlier. It is clear that the result will also be different if the operation is performed by several surgeons.

6. Line 149. If the operation time is long, the result is usually not good. Do you think the results are related to the time of surgery?

7. Line 213-214. I don't quite understand "the lack of guidance". Can you elaborate more for the readers?

Author Response

Thanks for the comments. The responses to your comments are as follows:

  1. In lines 39-46 and 54-58, the references and introduction highlighting the benefits of PFNs have been modified in bold.
  2. In the section on limitations, the wide age range of the patient population was highlighted in bold lines 342-343 as the most significant limitation of the study.
  3. After the surgical procedure, the follow-up was described in bold on lines 169–177.
  4. The mechanism of the V-effect is intended to be described in bold between lines 313 and 318.
  5. As mentioned in lines 136-137 in bold, the same surgical team, consisting of five surgeons with five years of experience in orthopaedic trauma, performed all surgeries.
  6. Lines 309–310 are bolded to highlight our belief that prolonged surgery may be at the root of the poor clinical outcomes seen in A-PFN patients.
  7. Between lines 291 and 294, the phrase "lack of guidance" was changed and tried to be made clearer.

Reviewer 3 Report

Dear authors,

An interesting topic is highlighted here. Three similar designs of Proximal Femoral Nails (PFN) with distinguishing differences and their application to a variety of patients are compared according to clinical , radiological and functional results. Readers are not familiar with nails used in Turkiye so a useful addition to would be a paragraph where differences of design and surgical technique are presented. However, my major concern is the lack of any evidence about comparability of the three study group. To be exact, according to what a specific nail was used to a specific patient. There is no info about allocation and concealment and no mention about the surgical team; was it the same for all operations or different? If there were different surgeons, were equally experienced? All the above in combination with the lack of any kind of power analysis and statistical analysis regarding the heterogeneity of groups constitute a major flaw of your study. 

Minor points are raised and I mention them below:

Line 12: Please describe which kind is your study: perspective, retrospective, etc.

Line 31: In a scientific paper there is no place of personal aspects and view in Conclusion section, only recommendations and conclusions which are supported by paper results.

Lines 51-52: Please clarify the meaning of the sentence; you mentioned numerous studies and your references are only 4. Please define the term "market".

Lines 61-62: The consent form was provided by patient preoperatively and regarded the kind of the PFN was used or the operation in general? Moreover, the approval of ethics committee was given in 2021, in which way this committee provided consent forms to patients in 2018?

Line 64: Do you mean "the follow-up period was at least 1 year" by saying "were observed etc.?"

Line 65: Bilateral fractures simultaneously or any pre-existed intertrochanteric fracture in the contralateral leg was an exclusion criterium?

Line 98: CDA , please explain this abbreviation.

Lines 129-130: Was partial weight bearing recommended to all patients without any exception regarding for instance comminuted fractures or highly osteoporotic bone? In case of polytrauma patients in Intensive Care Units which was the protocol of physiotherapy?

Line 143: As I mentioned in the first paragraph, a range of patient age 23 to 95 could cause major statistical errors which should be avoided.

Line 171 (Table 1): Please clarify these terms: "Preoperative day", "Transfusion", "Hospitalisation"

Line 175 (Table 2): Please explain how was evaluated the quality of reduction.  Moreover, in complications please clarify the terms "Pain" (When was a complication? After which post-operative day or month? Any pain - even a minor pain equal to disturbance- was evaluated?), "Varus" (Malalignment according to what?), "V-Effect" (A varus malalignment more than how many degrees was evaluated). Moreover, who was the surgeon who judged the clinical, functional and radiological results? A member of your team? An indepented orthopaedic surgeon? The same person who operated the patients?

Lines 200-201: Was absolutely necessary to use both lag screws in every fracture? In some cases and nail designs the second screw is optional according the distinctive nature of the fracture, the bone quality and the experience of the surgeon.

Line 237: Wrong reference number, please be careful. 

Line 246: There is no evidence (lack of power analysis) that your study has a sufficient sample size. 

Author Response

Thank you for the comments. The responses to your comments are as follows:

The study's introduction, methodology, and conclusion have all been improved, and any new words have been bolded for emphasis. Also, intended detailed description of PFN designs can be found in the PFN characteristics section, lines 106-132. Lines 137–148 have been bolded to emphasise some of the explanations that have been added to the section on surgical techniques. The parameters for determining the PFN type to use for a specific patient are bolded between lines 101 and 105. Information about the surgical team is indicated in bold on lines 136-137.

  1. As highlighted in bold in line 11, this is a retrospective study.
  2. Lines 30-32 have been modified and bolded.
  3. The terms 'market' and 'numerous' are modified and bolded on lines 57-58.
  4. Patients' written informed permission was sought for surgery in general regardless of the kind of PFN. In 2021, the ethical committee approved this retrospective study that would only depend on hospital records and computerised information systems, including the written consent of patients who had surgery in 2018.
  5. The sentence at line 71 has been altered and made bold.
  6. Exclusion criteria have been modified in lines 72–74.
  7. The term "CDA" is explained as the collodiaphyseal angle.
  8. Lines 158–164 provide definitions in boldface style that relate to partial weight bearing. The existence of polytrauma was one of the exclusion criteria listed in bold on lines 72 and 73. The postoperative rehabilitation of intensive care patients is described in bold between lines 159 and 163.
  9. In the section on limitations, the wide age range of the patient population was highlighted in bold lines 342-343 as the most significant limitation of the study.
  10. Table 1 has modified and bolded terms for ''Preoperative day, Transfusion, and Hospitalization''.
  11. How the quality of postoperative reduction is evaluated is highlighted in bold between lines 95 and 100. The determination of hip and thigh pain and the measurement of varus collapse are highlighted in bold between lines 172 and 178. The patients' radiological results including the V-effect and varus collapse were evaluated by an independent orthopedic surgeon as indicated in bold on lines 95-98. Clinical assessments were made by the same surgical team that performed the surgery, documented in patients' files, and shown in bold on lines 170-178.
  12. As a result of the nature of InterTAN and PROFIN nails, we were required to utilise double screws while installing them. We were limited in our options due to the fact that the implant offered in line with the hospital's purchase terms varied over time.
  13. The reference number is given appropriately and in bold at line 331.
  14. On lines 342-353, the study's limitations and strengths are highlighted and modified in bold.

Round 2

Reviewer 3 Report

Dear authors,

A clear improvement regarding quality and scientific soundness is detected throughout the manuscript. I really appreciate your effort. However, my major concern was not overcome; Are the three groups comparable? Is there any heterogeneity? A statistical specialist could help you using some extra tools and parameters to describe quality characteristics of patients. Your statistical analysis is basic and this reduces the quality of presentation. 

The fact that hospital management was responsible to choose the nail type for each and every patient enhances the quality of your retrospective study regarding blindness, concealment and allocation. These aspects should be mentioned. 

I am going to raise some minor concerns regarding specific sections.

Abstract and Introduction: Clear improvement. Nothing to add or change.

Materials and Methods: 

Lines 72-76: I would propose a phrasing like " Non-trochanteric fractures, pathologic fractures, polytrauma patients, bilateral simultaneous fractures, previous intertrochanteric fractures in the contralateral leg, impaired muscle strength (patients reduced mobility could not be an exclusion criterium as geriatric population is included), mental disorders (dementia included or excluded?) and follow-up less than a year are exclusion criteria."

Lines 101-105: Please make clear that hospital administration is charged for the type of nail. Neither doctors nor patients are responsible for this decision and are aware of the type of nail used just before the operation. 

Lines 159-162: A sedated patient in ICU is really unable to walk or even move. Please rephrase.

Lines 164-165: It is not clear what you mean. Is long surgical time a result of delayed closure?

Lines 172-173: Please provide quality features of pain; just a mild pain responded to paracetamol or a severe pain restricted patient's ability for independence? These different types should be classified accordingly.

Results: See above (statistical analysis and group quality analysis, heterogeneity tests)

Discussion and Conclusion: Nothing to comment or add. 

Author Response

Thank you for the comments. The responses to your comments are as follows:

The results of the Shapiro-Wilk test, which measures the conformity of the data to the normal distribution, are given in Table 1. The results of the Shapiro-Wilk test as well as the histogram graphics and Boxplot findings were considered while deciding which test to apply. Because of their length, histogram and Boxplot findings are not included in the final draft. Lines 181–185 and 194-196 have this data in boldface type. In addition, the statistics tables section was improved. Tables 5 and 7 describe the results of post hoc analyses, which included a more in-depth look at the variables responsible for the difference. Moreover;

  1. Lines 72-76, modified and bolded as indicated by the reviewer.
  2. On lines 102-104, the situation is stated more clearly and with bolder font.
  3. Between lines 158 and 160, the phrase has been modified in bold.
  4. Further detail has been added to the description of the phrase found between lines 163 and 165.
  5. The referee-requested modifications to the description of pain are made in bold between lines 171 and 175.
  6. The article has been updated with the addition of three tables and a reorganisation of the order in which those tables are referenced.

Round 3

Reviewer 3 Report

Dear authors,

Every point raised previously was addressed with scientific soundness. I have nothing to comment further.